

# Effects of different ratios of nitrogen base fertilizer to topdressing on soil nitrogen form and enzyme activity in sugar beet under shallow drip irrigation

Zhi Li[1], Caiyuan Jian[1], Xiaoxia Guo[1], Lu Tian[1], Kang Han[1], Yinghao Li[1], Peng Zhang[2], Dejuan Kong[2], Huimin Ren[1], Alehesi Jiaerdemulati[3], Zhenzhen Wang[2], Huiyu Liu[2], Chunyan Huang[1] and Wenbin Su[1]

[1] Special Crops Institute, Inner Mongolia Academy of Agricultural & Animal Husbandry Sciences, Hohhot, Inner Mongolia, China
[2] Ulanqab Institute of Agriculture and Forestry Science, Jining, Inner Mongolia, China
[3] Xinjiang Ili Kazakh Autonomous Prefecture Agricultural Science Research Institute, Ili, Xinjiang, China

## ABSTRACT

Sugar beets account for 30% of global sugar production each year, and their byproducts are an important source of bioethanol and animal feed. Sugar beet is an important cash crop in Inner Mongolia, China. To achieve high yields and sugar content, it is essential to supply nitrogen fertilizer in accordance with the growth characteristics of sugar beet, thereby enhancing the efficiency of nitrogen fertilizer utilization. A two-year experiment was carried out in the experimental field of the Inner Mongolia Academy of Agricultural & Animal Husbandry Sciences. The impact of varying ratios of nitrogen-based fertilizer to topdressing on nitrate nitrogen and ammonium nitrogen levels in the 20–60 cm soil layer, as well as the activities of protease, urease, catalase, and sucrose in the 20–40 cm soil layer were investigated during the rapid leaf growth period and root and sugar growth period. Results indicated that different ratios of nitrogen-based fertilizer to topdressing significantly influenced the levels of nitrate nitrogen and ammonium nitrogen, and the activities of protease and urease in the 0–20 cm soil layer, with these effects diminishing as soil depth increased. The activities of catalase and sucrose were minimally impacted. Nitrogen was applied at 150 kg/ha during the growth period of sugar beet, according to the growth characteristics of sugar beet to maximize nitrogen utilization efficiency. Topdressing was completed with irrigation at the rapid growth stage. The nitrogen-based fertilizer to topdressing ratio of 6:4 resulted in optimal crop yield and sugar yield of sugar beet under shallow drip irrigation. Additionally, the activities of protease and urease in different soil treatments were significantly different, and the activities of protease and urease in the 0–40 cm soil layer were identified as useful soil physiological indicators for nitrogen utilization in sugar beet.

Corresponding authors
Chunyan Huang, hcy86@aliyun.com
Wenbin Su, swb1964@sina.com

# INTRODUCTION

Sugar beet is a crucial raw material of sugar and serves as a significant economic crop in arid and cold regions of China (*Jammer et al., 2020*). In 2022 and 2023, the sugar beet planting area in Inner Mongolia was 12 and 9.33 hectares, respectively, accounting for 60.2% and 54.45% of the total sugar beet planting area in China, respectively. Shallow drip irrigation is considered an advanced technology in irrigation, capable of enhancing soil enzyme activity to ultimately boost sugar beet yield and sugar content (*Li et al., 2024*). Sugar beet, with its high biological yield, requires a substantial amount of fertilizer (*Hergert, 2011*; *Stevanato et al., 2015*). A well-designed irrigation and fertilization system plays a crucial role in fostering the healthy, environmentally friendly, and sustainable growth of the sugar beet industry in Inner Mongolia.

Nitrogen fertilizer significantly influences the yield and sugar content of sugar beets (*Li et al., 2023b*; *Wolfgang et al., 2023*). Unlike other crops, sugar beets require large amounts of nitrogen fertilizer to achieve high yields. Nitrate nitrogen and ammonium nitrogen are two common forms of nitrogen found in soil, and can reflect the nitrogen utilization of plants. The rapid growth period of leaves is the fastest above-ground growing stage of sugar beet and the growth period of roots and sugar is the fastest growing stage of the underground parts of sugar beet (*Li et al., 2019*).

Soil nutrients and enzymes play a crucial role in crop growth, with their synergistic effect enhancing crop nitrogen efficiency (*Sun et al., 2023*; *Yang et al., 2022*). Soil enzymes originate mainly from microorganisms and plant roots, as well as the enzymes released from plant and animal residues. Soil enzyme activity serves as a key indicator influencing soil nutrient cycling and provides an objective assessment of soil nutrient levels (*Lu et al., 2023*; *Han et al., 2023*; *Boros-Lajszner, Wyszkowska & Kucharski, 2023*). Application of nitrogen fertilizer can enhance soil enzyme activities. However, reducing nitrogen fertilizer dosage appropriately can boost soil protease and urease activities, thus regulating soil nitrogen transformation and facilitating nitrogen absorption by crops in the soil (*Fu et al., 2017*). The activities of sucrose and urease, as well as maize yield, have also been shown to increase with nitrogen fertilizer application (*Bai et al., 2022*). A previous study found that under saline-alkali stress, the enzyme activity of sugar beet soil significantly increased with nitrogen reduction (*Lu et al., 2022*). Another study found an optimal amount of nitrogen fertilizer during the early growth stage of sugar beet can promote above-ground growth and provide a necessary 'source' for root development and sugar accumulation (*Li et al., 2019*).

This study investigated the impact of soil nitrogen forms and soil enzyme activity on yield, sugar content, and nitrogen fertilizer efficiency in sugar beet cultivation. Some soil physiological indexes were found to maximize the nitrogen use efficiency of sugar beet. These results provide a theoretical basis for rational fertilization and improvement of nitrogen use efficiency in sugar beet production.

**Table 1  Soil nutrient status.**

| Years | Total nitrogen mg/kg | Alkali-hydrolyzed nitrogen mg/kg | Available phosphorus mg/kg | Available potassium mg/kg | Organic Matter mg/g | Organic carbon mg/g | pH |
|---|---|---|---|---|---|---|---|
| 2020 | 290.22 | 85.35 | 53.78 | 187.2 | 27.33 | 12.56 | 8.05 |
| 2021 | 275.39 | 74.67 | 37.60 | 158.1 | 20.26 | 13.35 | 7.75 |

## MATERIALS & METHODS

### Site description

Field trials were performed in Hohhot city, Inner Mongolia, China (N40°46′22, E111°39′), at an altitude of 1,000 m. The average temperature in this area from April to October was 16 °C, and the total rainfall was 450 mm. The climatic conditions in this study were classified as temperate continental monsoon climate (*Zhou et al., 2022a*). The experiment was conducted in the experimental field of Inner Mongolia Academy of Agricultural & Animal Husbandry Sciences, with the soil being primarily clay loam. The specific soil nutrient content is shown in Table 1.

### Material

The single seed sugar beet variety IM1162, imported from the Dutch Ande Company, was used in this experiment.

### Experimental design

Sugar beets were sown on April 23, 2020 and April 25, 2021. Drip irrigation was employed, with a belt placed between two rows of sugar beets. The drip-irrigation belt was buried at a shallow depth of three cm. Throughout the growth period, the sugar beets were irrigated four times, each time with 450 m$^3$/ha of water and a total nitrogen application of 150 kg/ha. Irrigation was performed at the seedling stage, rapid leaf growth stage, and root and sugar growth stage, and nitrogen fertilizer was applied once at the rapid leaf growth stage. Sugar beet cultivation focused on harvesting roots, with nitrogen fertilizer primarily used as a base fertilizer. Four treatments were established based on different ratios of base fertilizer to topdressing: N1 (8:2), N2 (7:3), N3 (6:4), and N4 (5:5). The total nitrogen fertilizer applied as a base was considered the control (N0, 10:0). The plots measured 15 m × 5 m, with an inter-row measurement of 50 cm and an intra-row measurement of 23 cm, resulting in a total area of 75 m$^2$ for each plot. The theoretical number of plants per hectare was calculated to be 87,000. Nitrogen fertilizer in the form of urea (46% nitrogen content), phosphate fertilizer in the form of heavy superphosphate (46% P$_2$O$_5$), and potassium fertilizer in the form of potassium sulfate (50% K$_2$O) were used. The phosphate and potassium fertilizers were applied once as base fertilizer, while the nitrogen fertilizer was applied as topdressing during the rapid leaf growth period using drip irrigation. The experimental field management followed standard field planting patterns.

### Measurements and calculations

Soil samples were taken from the 0–20 cm, 20–40 cm, and 40–60 cm layers during both the rapid leaf growth period and the root and sugar growth period of sugar beet and were

stored at −20 °C. Soil samples were then used to analyze soil nitrate nitrogen, ammonium nitrogen, and soil enzyme activities.

Soil nitrate nitrogen was quantified using the phenol disulfonic acid colorimetric method, while soil ammonium nitrogen was determined using the indophenol blue colorimetric method (*Gong et al., 2020*). Soil protease activity was assessed *via* copper salt colorimetry. Urease activity was measured using the sodium phenol-sodium hypochlorite colorimetric method, and sucrose activity was determined by resorcine-resorcinol colorimetry. Additionally, catalase activity was determined using the dinitrosalicylic acid colorimetric method (*Gong et al., 2020*).

Sugar beets were harvested in the two growing years on October 5, 2020 and October 7, 2021, respectively. During sugar beet harvesting, the roots were harvested using a one-sizer method. The middle 5 m$^2$ of each plot was selected to measure crop yield, with all sugar beets in that area weighed using electronic scales. The brix value was determined with a Nissan Refractometer PAL-1 Brix meter. The sugar yield (kg/ha) = yield (kg/ha) × sugar content (%). Sugar content (%) = Brix value × 0.8.

## Data analysis

The data were analyzed by ANOVA with the SAS 9.0 statistical software package (SAS Institute, Cary, NC, USA). Duncan's multiple range test ($P < 0.05$) was used to determine the statistically significant differences. Data processing was conducted using Microsoft Excel 2007, and graphs were created with GraphPad Prism 5.0 software and Origin 2021.

## RESULTS

### Effects of different ratios of nitrogen base fertilizer to topdressing on soil nitrogen form

The nitrate nitrogen content in the 0–20 cm soil layer was significantly influenced by the ratio of nitrogen base fertilizer to topdressing, with a higher nitrate nitrogen content observed as the ratio increased (Fig. 1). During the root and sugar growth period, the nitrate nitrogen content of N3 and N4 was 31.82% and 51.38% higher than that of N0, respectively. However, the ratio of nitrogen base fertilizer to topdressing had minimal impact on the nitrate nitrogen content in the 20–60 cm soil layer.

Ammonium nitrogen and nitrate nitrogen content was consistent during the rapid leaf growth period. In the root and sugar growth periods, as the ratio increased, the ammonium nitrogen in the 0–20 cm soil layer exhibited an upward trend. Specifically, the ammonium nitrogen content of N3 and N4 was 40.89% and 49.89% higher, respectively, than that of N0 in the rapid leaf growth period. However, there was no significant difference observed with the total base application of nitrogen fertilizer. The effect of ammonium nitrogen on the 40–60 cm soil layer was minimal.

### Effects of different ratios of nitrogen base fertilizer to topdressing on soil enzyme activity

Soil protease serves as a crucial indicator of nitrogen availability within the soil profile. During the rapid leaf growth period, the soil protease activity in the 0–20 cm soil layer

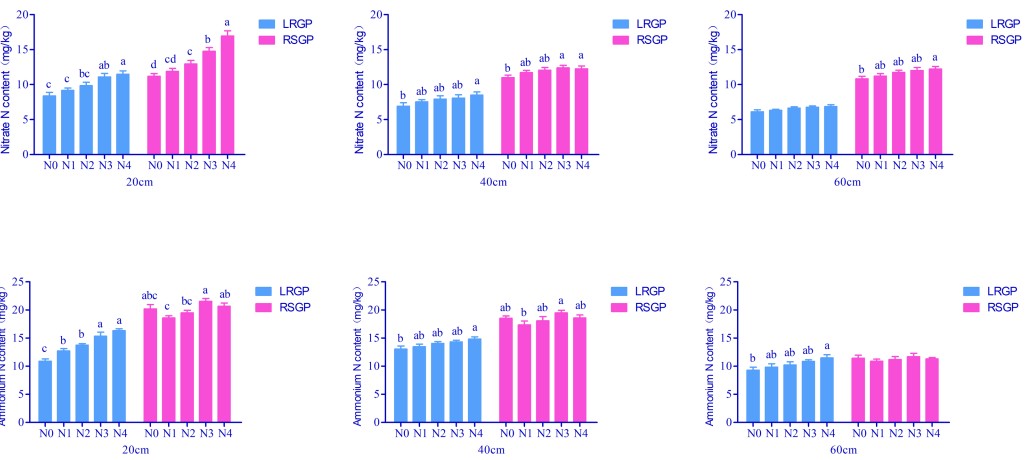

**Figure 1** **Effects of different ratios of base fertilizer to topdressing on soil nitrate nitrogen and ammonium nitrogen.** Different lowercase and uppercase letters indicate significant ($P < 0.05$) differences. LRGP: The rapid leaf growth period. RSGP: The root and sugar growth period.

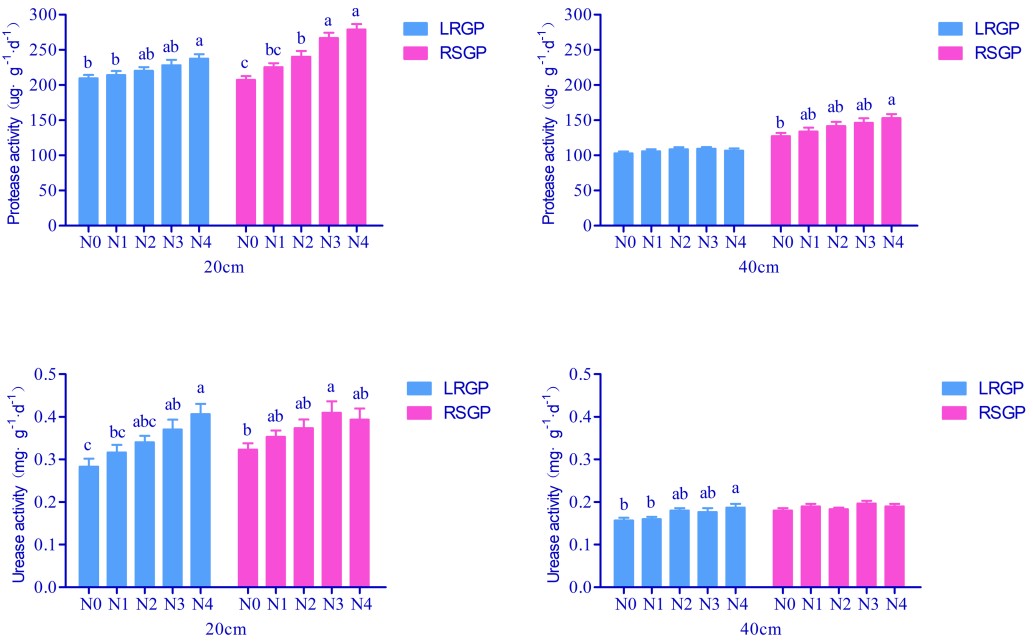

**Figure 2** **Effects of different ratios of base fertilizer to topdressing on soil protease and urease activities.** Different lowercase and uppercase letters indicate significant ($P < 0.05$) differences. LRGP: The rapid leaf growth period. RSGP: The root and sugar growth period.

treated with N4 was notably higher than in the 0–20 cm soil layer treated with N0 and N1 (Fig. 2). Conversely, in the root and sugar growth period, the soil protease activity in the 0–20 cm soil layer increased significantly as the ratios increased. Specifically, the soil protease activity of N3 and N4 was 28.67% and 34.48% higher, respectively, than that of N0 during the root and sugar growth period.

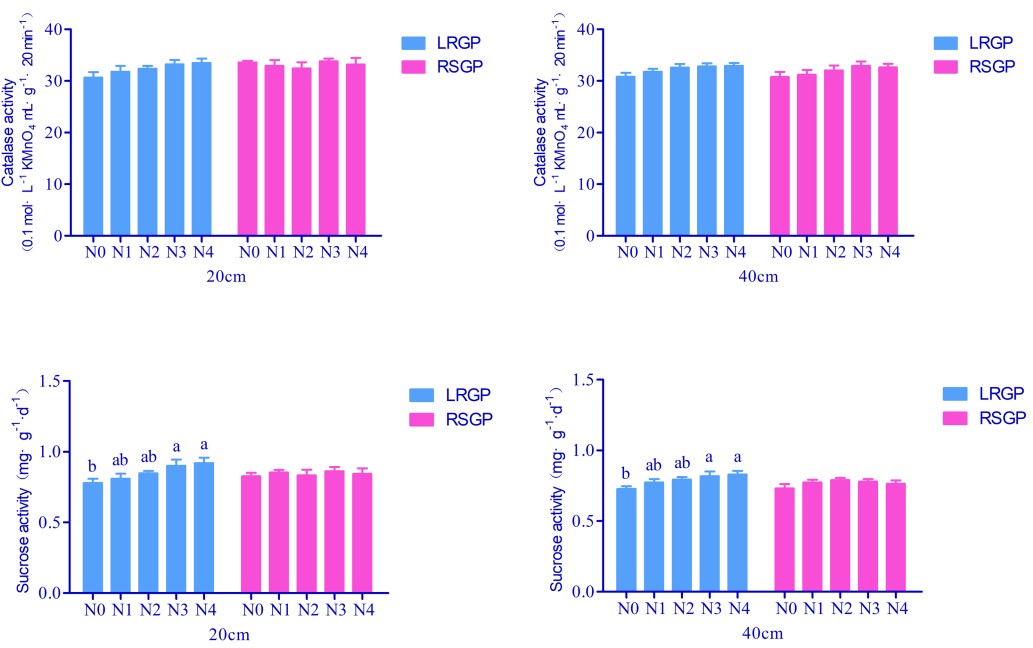

**Figure 3** Effects of different ratios of base fertilizer to topdressing on soil catalase and sucrase activities.

Soil urease is an important factor reflecting soil nitrogen supply capacity. In the 0–20 cm soil layer, urease activity was notably affected by the ratio of nitrogen base fertilizer to topdressing during the rapid leaf growth period and root and sugar growth period. During the rapid leaf growth period, the urease activity of N3 and N4 was 31.07% and 42.84% higher, respectively, than that of N0 (Fig. 2).

Soil catalase activity is a crucial indicator of plant stress resistance. This experiment found no significant difference in catalase activity among different soil treatments during two distinct growth periods (Fig. 3).

Soil sucrose activity plays a vital role in the soil carbon cycle. Notably, in the 0–20 cm and 20–40 cm soil layers during the rapid lead growth period, soil treated with N3 and N4 exhibited significantly higher sucrose activity than the soil treated with N0. Specifically, sucrose activity was 15.73% and 18.01% higher in the 0-20 cm layer for N3 and N4, respectively, than in N0 (Fig. 3). No significant differences were observed among the other treatments.

## Effects of different ratios of nitrogen base fertilizer to topdressing on yield, sugar content, and agronomic nitrogen use efficiency ($AE_N$) of sugar beet

The results from this two-year experiment demonstrated a positive correlation between nitrogen base fertilizer to topdressing ratio and sugar beet yield. In 2020, the yields of N1, N2, N3, and N4 were 3.01%, 4.91%, 7.86%, and 6.09% higher than N0, respectively (Table 2). Among these, the N3 treatment exhibited the highest yield. The sugar content

**Table 2  Effects of different base to topdressing ratio of nitrogen fertilizer on yield, sugar content of sugar beet, and AE$_N$.**

| Years | Treatments | Yield kg/ha | Sugar content % | Sugar yield kg/ha | AE$_N$ |
|---|---|---|---|---|---|
| | N0 | 68,046.08 ± 1,898.57 c | 16.40 ± 0.06 a | 11,159.95 ± 339.52 b | 81.37 ± 12.66 c |
| | N1 | 70,093.48 ± 1,293.17 bc | 16.34 ± 0.07 a | 11,455.62 ± 184.37 ab | 95.02 ± 8.62 bc |
| 2020 | N2 | 71,386.86 ± 1,441.72 ab | 16.30 ± 0.08 ab | 11,634.35 ± 214.58 ab | 103.65 ± 9.61 ab |
| | N3 | 73,396.34 ± 1,504.67 a | 16.25 ± 0.08 ab | 11,925.84 ± 258.16 a | 117.04 ± 10.03 a |
| | N4 | 72,186.83 ± 1,663.85 ab | 16.17 ± 0.09 b | 11,673.09 ± 327.93 ab | 108.98 ± 11.09 ab |
| | N0 | 64,765.98 ± 1,490.61 c | 16.19 ± 0.13 a | 10,483.88 ± 312.72 c | 74.96 ± 9.94 c |
| | N1 | 66,569.96 ± 982.35 bc | 16.26 ± 0.10 a | 10,826.16 ± 134.66 bc | 86.99 ± 6.55 bc |
| 2021 | N2 | 68,449.75 ± 1,762.12 ab | 16.38 ± 0.13 a | 11,211.02 ± 244.90 ab | 99.52 ± 11.75 ab |
| | N3 | 69,970.08 ± 1,617.84 a | 16.32 ± 0.09 a | 11,421.20 ± 299.78 a | 109.66 ± 10.79 a |
| | N4 | 68,069.90 ± 1,585.85 ab | 16.23 ± 0.11 a | 11,045.25 ± 187.79 ab | 96.99 ± 10.57 ab |

**Notes.**

AE$_N$, Agronomic nitrogen use efficiency.
Different lowercase and uppercase letters indicate significant ($P < 0.05$) differences.

decreased as the nitrogen base fertilizer to topdressing ratio increased, with the lowest sugar content observed in the N4 treatment. Additionally, the sugar yield of N3 was significantly higher than that of N0, showing a 6.86% increase. There were no significant differences between the other treatments. N3 also had the highest AE$_N$, with significant increases of 43.83% and 23.17% compared to N0 and N1, respectively. These findings were consistent between field trials conducted in 2020 and 2021.

## Correlation of crop yield and sugar content of sugar beet with soil nitrogen form and enzyme activity

A correlation analysis revealed a strong positive correlation between the crop yield of sugar beet and both sugar yield and soil urease activity (Fig. 4). Additionally, there was a significant positive correlation between sugar beet yield and soil protease activity. Conversely, the sugar content of sugar beet showed a strong negative correlation with soil nitrate nitrogen content and soil protease activity. Furthermore, a strong positive correlation was found between sugar yield and soil urease activity. This study also identified a significant positive correlation between soil nitrate and protease activity, as well as a positive correlation between protease and urease activity in the soil.

## DISCUSSION

Nitrogen fertilizer affects the yield and sugar content of sugar beet. Excessive nitrogen application during the later stages of sugar beet growth can disrupt the transition of sugars within the plant, impacting both sugar beet yield and sugar content. Previous studies showed that the amount of nitrogen applied during the growth period of sugar beet was 150 kg/ha (*Li et al., 2019*). This study sought to identify the best way to apply nitrogen fertilizer according to the growth period of sugar beet to maximize nitrogen use efficiency. Since nitrogen fertilizer can increase the yield of sugar beet, but also reduce the
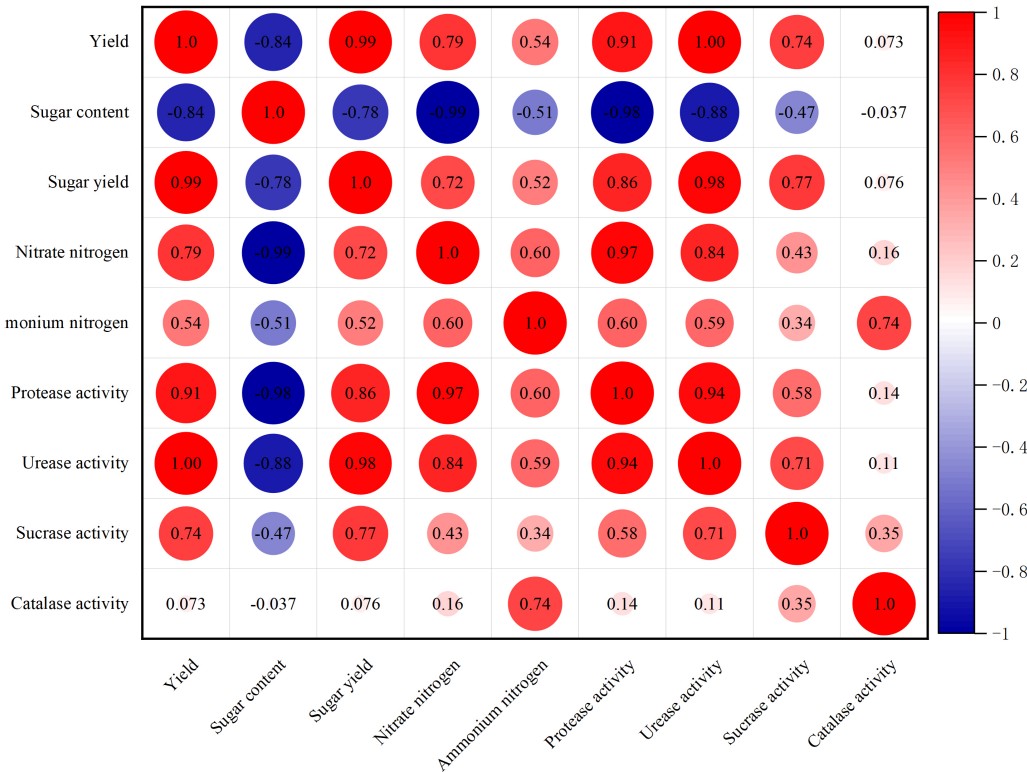

**Figure 4** Correlation of crop yield and sugar content of sugar beet with soil nitrogen form and enzyme activity.

sugar content, sugar yield was used to determine the optimal nitrogen base fertilizer to topdressing ratio.

Fertilization can significantly affect the enzyme activity in soil (*Zhao et al., 2022*). Enzymes are involved in the development of soil nutrients and are biologically active proteins in the soil (*Li et al., 2023a*; *Dong et al., 2024*). Soil enzyme activity can vary significantly depending on soil texture and the specific crop species (*Song et al., 2023*). The impact of nitrogen application on soil enzyme activity also varies by soil type (*Ullah et al., 2023a*). Soil enzyme activity plays a critical role in influencing soil microbial activity (*Qu et al., 2023*). The application of organic fertilizer and microbial fertilizer has been shown to increase the activities of sucrose, urease, and protease in the soil (*Huang et al., 2023*). Soil protease plays a key role in converting amino acids, proteins, and other nitrogen-containing organic compounds in the soil, with its hydrolyzed products serving as nitrogen sources for plants. This study revealed a significant increase in soil protease levels with higher nitrogen base fertilizer to topdressing ratios during the root and sugar growth periods.

Soil urease is a crucial indicator of soil nitrogen supply capacity, with its activity directly impacting the nitrogen balance in soil. Research by *Tahir et al. (2023)* and *Singh et al. (2023)* found that urease activity increased significantly with higher nitrogen application rates. During wheat growth, soil urease activity initially decreased before rising again, reaching its lowest point at the jointing stage. Conversely, soil sucrose activity, $NH_4^+$-N,

and $NO_3^--N$ contents followed a pattern of increase followed by decrease, peaking at the flowering stage (*Gong et al., 2020*). This research demonstrated a significant increase in soil urease levels with higher nitrogen base fertilizer to topdressing ratios during the rapid leaf growth phase of sugar beet.

Soil sucrose plays an important role in decomposing sucrose in soil into small molecular monosaccharides and facilitating the conversion of sucrose, thus influencing the soil carbon cycle (*Ji et al., 2024*). This experiment demonstrated that the ratio of nitrogen base fertilizer to topdressing did not significantly impact soil sucrose activity.

Soil catalase is of great significance to the study of dryland crops because it breaks down hydrogen peroxide into water and hydrogen through enzymatic reactions, thus reducing the harmful effects of hydrogen peroxide on organisms, improving crop stress resistance. A study by *Feng et al. (2023)* found that tomato rotation could notably boost soil catalase activity. However, in a rice experiment by *Ullah et al. (2023b)*, the effect of nitrogen application on soil catalase was found to be insignificant. This aligns with the findings of the present study, which also observed no relationship between soil catalase levels and increased nitrogen base fertilizer to topdressing ratios.

Nitrogen available to plants can be categorized into organic nitrogen and inorganic nitrogen, with ammonium nitrogen and nitrate nitrogen being the primary forms of inorganic nitrogen taken up by roots (*Liu et al., 2024*; *Li et al., 2022*; *Bao et al., 2024*; *Lei et al., 2024*). Enzymes such as nitrate reductase, glutamine synthetase, and glutamate synthase play crucial roles in plant nitrogen absorption and utilization. Nitrate ($NO_3^--N$) is converted to ammonium ($NH_4^+-N$) by NR and NiR and then further assimilated into amide nitrogen by enzymes like GOGAT and GS before being absorbed and used by crops (*Zhang et al., 2022*; *Meng et al., 2024*). As fertilizer application increased, soil nitrate nitrogen and ammonium nitrogen contents also increased. In a previous study, this led to an initial increase in potato yield and soil enzyme activity, followed by a decrease (*Zhang et al., 2023*). Under sugarcane nitrogen treatment, the levels of nitrate nitrogen and ammonium nitrogen in the soil decreased with greater soil depth. Nitrate and ammonium nitrogen accumulation was generally low in the 0–10 cm soil layer, but high in the 20–40 cm layer (*Mao et al., 2023*). Another study found soil nitrate nitrogen levels in the 0–40 cm soil layer decreased for corn (*Zhang et al., 2018*). *Zhou et al. (2022a)* and *Zhou et al. (2022b)* found that the concentrations of ammonium nitrogen and nitrate nitrogen in the 0–20 cm rhizosphere soil of highland barley were 1.76 times and 2.00 times higher than those in the 20–40 cm soil layer, respectively, suggesting that the conversion, absorption and transport of substances primarily occur in the upper soil. Additionally, once the concentration of ammonium nitrogen in the soil reached a certain level, there was no significant correlation between the concentration of nitrate nitrogen and ammonium nitrogen (*Zhou et al., 2022b*). The findings of the present study are consistent with previous research findings, indicating a decrease in soil nitrate nitrogen and ammonium nitrogen levels in the 0–60 cm soil layer of sugar beet as soil depth increased. The highest enzyme activities were observed in the 0–20 cm soil layer. Additionally, soil nitrate nitrogen levels were found to increase with the ratio of nitrogen base fertilizer to topdressing. Furthermore, no significant

correlation was found between nitrate nitrogen and ammonium nitrogen content in the soil.

## CONCLUSIONS

This study showed that under shallow drip irrigation, the optimal ratio of nitrogen base fertilizer to topdressing was 6:4. Topdressing was completed with irrigation at the rapid growth stage, resulting in maximum crop yield and sugar yield of sugar beet. Additionally, the activities of protease and urease differed significantly with different soil treatments, and the activities of protease and urease in the 0–40 cm soil layer were identified as useful soil physiological indicators for nitrogen utilization in sugar beet. In the future, we hope to study the relationship among soil microorganisms, soil enzyme activity with yield and sugar content of sugar beet.

## ACKNOWLEDGEMENTS

We received a great deal of support and assistance throughout this study. We would particularly like to acknowledge the cultivation physiology of sugar beet research team for their collaboration and support.

### Funding

This work was supported by the China Agriculture Research System of MOF and MARA (CARS-17), the Inner Mongolia Academy of Agricultural and Animal Husbandry Sciences Youth Innovation Fund (2021QNJJNO6) and the Science and technology planning project of Inner Mongolia (2022YFDZ0065). The funders had no role in study design, data collection and analysis, decision to publish, or preparation of the manuscript.

### Grant Disclosures

The following grant information was disclosed by the authors:
China Agriculture Research System of MOF and MARA: CARS-17.
Inner Mongolia Academy of Agricultural and Animal Husbandry Sciences Youth Innovation Fund: 2021QNJJNO6.
Science and technology planning project of Inner Mongolia: 2022YFDZ0065.

### Competing Interests

The authors declare there are no competing interests.

### Author Contributions

- Zhi Li conceived and designed the experiments, performed the experiments, analyzed the data, prepared figures and/or tables, authored or reviewed drafts of the article, and approved the final draft.
- Caiyuan Jian performed the experiments, authored or reviewed drafts of the article, and approved the final draft.

- Xiaoxia Guo performed the experiments, authored or reviewed drafts of the article, and approved the final draft.
- Lu Tian performed the experiments, authored or reviewed drafts of the article, and approved the final draft.
- Kang Han performed the experiments, authored or reviewed drafts of the article, and approved the final draft.
- Yinghao Li analyzed the data, prepared figures and/or tables, and approved the final draft.
- Peng Zhang analyzed the data, prepared figures and/or tables, and approved the final draft.
- Dejuan Kong analyzed the data, prepared figures and/or tables, and approved the final draft.
- Huimin Ren analyzed the data, prepared figures and/or tables, and approved the final draft.
- Alehesi Jiaerdemulati analyzed the data, prepared figures and/or tables, and approved the final draft.
- Zhenzhen Wang analyzed the data, prepared figures and/or tables, and approved the final draft.
- Huiyu Liu analyzed the data, prepared figures and/or tables, and approved the final draft.
- Chunyan Huang conceived and designed the experiments, authored or reviewed drafts of the article, and approved the final draft.
- Wenbin Su conceived and designed the experiments, authored or reviewed drafts of the article, and approved the final draft.

## Data Availability

The raw measurements are available in the Supplemental Files.

## Supplemental Information

Supplemental information for this article can be found online at http://dx.doi.org/10.7717/peerj.18219#supplemental-information.

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
