# Peer review of "Effects of different ratios of nitrogen base fertilizer to topdressing on soil nitrogen form and enzyme activity in sugar beet under shallow drip irrigation"

_PeerJ, doi:10.7717/peerj.18219_

## Round 0.1 · original submission · Major Revisions

Make your corrections by taking the reviewer reports into consideration.

Reviewer 1 ·

Basic reporting

*English grammar and writing must be revised.
*The article is supported by sufficient and up-to-date literature. References to previous studies are sufficient.
*The article was prepared in format. But if there is no difference between the data in the figures, I consider it unnecessary to give letters.
*The manuscript is self-consistent, presents an appropriate data set and contains all the results relevant to the hypothesis.

Experimental design

*This study is original research within the aims and scope of the journal.
*The article partly defines the research question, which should be relevant and meaningful. The knowledge gap being investigated is not identified and there is no explanation of how the study contributes to filling this gap.
*There is little mention of findings that indicate that the research was conducted rigorously and to a high technical standard. The research must have been conducted in accordance with the ethical standards of the field.
*The methods in the study were mentioned superficially and not explained with sufficient information.

Validity of the findings

*The impact of the study, its innovative approach, and its differences from previous studies have not been fully demonstrated.
*The data on which the conclusions are based are statistically reliable and controlled.
*The results are generally well expressed, but the conclusion is not sufficiently descriptive.

Additional comments

Effects of different base to topdressing ratio of nitrogen fertilizer on soil nitrogen form and enzyme activity in sugar beet under shallow drip irrigation (#100002)

In this study, the effects of varying base to topdressing ratios of nitrogen fertilizer on soil nitrogen forms and enzyme activity in sugar beet under shallow drip irrigation were investigated. The experimental design was appropriate, and the results are presented tables and figures. Nevertheless, the manuscript needs minor revision. First of all, English grammar and writing must be revised. I have listed below the points that I believe would make the article stronger if corrected.

Abstract

L32: ‘sugar yield of sugar beet’ should be changed ‘sugar yield for sugar beet’.

Introduction

I believe it would be beneficial for the reader to have a more comprehensive understanding of shallow drip irrigation. The literature should support the effects of this practice on sugar beets, particularly in terms of yield and sugar content. The difference between this practice and drip irrigation should be revealed.

L45: ‘China respectively’ should be changed ‘China, respectively’.
L47: ‘Sugar beet with high biological yield’ should be changed ‘Sugar beet, with its high biological yield’.
L45: ‘theoretical’ should be changed ‘a theoretical’.

Materials and Methods

Site description
L68-69: ‘The climatic conditions in this study are categorised as temperate continental monsoon climate.’ this sentence should be cited.
L71: For the soil properties presented in Table 1, the soil depth (0-20, 20-40 or 40-60 cm) should be reported.

Experimental design

It should be written when the sugar beet was sown.
I recommend using ‘ha’ instead of ‘hm2’.

L77-78: It should be written in detail in which periods irrigation and fertilising applications were made, how many times fertilisation was done.
L79-82: Treatments should be written more descriptively. For example, what do 8 and 2 represent in 8:2? In which development periods were they applied?
L82-83: ‘row spacing’ and ‘plant spacing’ should be replaced by ‘inter-row’ and ‘intra-row’.

Measurements and calculations
L92: Please explain in detail the LRGP and RSGP periods. With a reference, explain when these periods started and ended.
L100: It should be written when the sugar beet was harvested.

Results

Please remove the lettering (e.g. they are all ‘a’) from the figures if there is no difference in your data.
L116: ‘with higher’ should be changed ‘with a higher’.
L117: ‘with increasing’ should be changed ‘with an increasing’.
L132-134: ‘root and sugar growth period’ should be changed ‘root and sugar growth periods’.
L149-150: ‘sugar content and agronomic’ should be changed ‘sugar content, and agronomic’.
L153: ‘The N3’ should be changed ‘the N3’.

Discussion

All ‘sucrase’ should be changed to ‘sucrose’.

L173: It would be better to replace ‘significantly influenced’ with increasing or decreasing values.
L177: References should be given at the end of the sentence.
L188: ‘played’ should be changed ‘plays’.
L191: ‘root and sugar growth period’ should be changed ‘root and sugar growth periods’.
L204: ‘by breaking’ should be changed ‘because it breaks’.

Conclusion

The conclusion section is too short. I believe that the article will be stronger if this section is written in more detail in order to provide information for future studies.

Reviewer 2 ·

Basic reporting

The manuscript “Effects of different base to topdressing ratio of nitrogen fertilizer on soil nitrogen form and enzyme activity in sugar beet under shallow drip irrigation” deals with a comprehensive analysis of the impact of nitrogen fertilizer application ratios on the soil nitrogen levels and enzyme activities during critical growth stages of sugar beet. The objective of optimizing yield and sugar content through appropriate nitrogen management is highly relevant, considering the economic importance of sugar beet as a cash crop in Inner Mongolia.
Comments
• The study should provide more background on why the specific nitrogen base fertilizer to top dressing ratio of 6:4 was hypothesized to be optimal. I appreciate the author providing data that will be useful for fertilizer management.
• Information on the initial soil characteristics (e.g., organic matter content, pH, texture) before the experiment started is missing, which is crucial for understanding nutrient dynamics. If available please provide it.
• How was the nitrogen base fertilizer to topdressing ratio of 6:4 determined initially, and what was the basis for selecting this specific ratio range for the experiment?
• The study does not mention the environmental conditions (e.g., temperature, rainfall) during the experiment, which can significantly influence results.
• What statistical methods were used to analyze the data, and were these methods appropriate for the study design? Were the sample sizes for each treatment group adequate to ensure robust statistical power?
• There is no discussion on the potential long-term impacts of different nitrogen fertilization strategies on soil health and sustainability. Kindly discuss.
• The study should discuss why effects diminished with soil depth and whether deeper soil layers were considered.
• Were there any observed differences in soil microbial communities between the different nitrogen treatments, and if so, how might these have influenced enzyme activities?
• Details on the statistical methods used to analyze the data are missing, making it difficult to evaluate the robustness of the conclusions.
• How were potential confounding variables (e.g., pest pressure, disease incidence) controlled for during the experiment? What were the economic implications of using the 6:4 nitrogen ratio compared to other ratios or fertilization methods?
• Minimal impacts on catalase and sucrase activities suggest that these enzymes are less responsive to nitrogen application methods, potentially due to their broader roles in soil organic matter decomposition and sugar metabolism.
• The study should provide clear, practical recommendations for farmers based on the findings in the conclusion section.

Experimental design

Please see section 1, Basic reporting

Validity of the findings

Please see section 1, Basic reporting

Additional comments

Please see section 1, Basic reporting

Reviewer 3 ·

Basic reporting

The article was written in a clear manner, with technically correct text, proper grammar, and accurate spellings. The figures and tables were well presented and structured, contributing positively to the overall readability. However, the article lacked references for several statements, and sufficient background information was not provided. Additionally, no clear hypotheses were presented, and the importance and novelty of the work were not highlighted, which would have added significant value to the content.

Experimental design

This original primary research documents the results of varying ratios of base to topdressing nitrogen fertilizer compared to a control with only base fertilizer applied. While the research question is well defined, the number of replications is not documented. Unfortunately, the knowledge gaps and overall contribution of the study were not mentioned. Additionally, the methods section lacks detail regarding sample collection, sample size, sampling approach, and data analyses. Procedures referred to as standards were not explained in detail, which hinders the reproducibility and clarity of the study.

Validity of the findings

The impact and novelty of the study were not assessed. Furthermore, the results of the statistical tests were inadequately reported, with p-values and post-hoc test results missing. The authors should report their results separately for the different plant growth periods to provide a clearer understanding of the impact of the varying nitrogen fertilizer ratios. The results are presented as loose observations linked to the original research questions, but without adequate statistical robustness, it is difficult to rely on such conclusions. Additionally, the discussion lacks depth in explaining the results and placing them in the context of past findings by other authors, as well as potential mechanisms behind the observed effects.
The conclusions were essentially a repetition of the results, failing to provide implications for management, highlight the importance of the findings, or offer an outlook for future research. This limited the potential impact and practical application of the study’s outcomes.

Additional comments

The authors present the results of an experiment investigating the impact of varying ratios of nitrogen base fertilizer to topdressing on nitrate nitrogen and ammonium nitrogen levels in the 20-60cm soil layer, as well as the activities of protease, urease, catalase, and sucrase in the 20-40cm soil layer during the leaf rapid growth and root and sugar growth periods. The manuscript should be thoroughly revised to include far more relevant background information, explaining the mechanisms behind nitrogen assimilation in sugar beet crops and its nutritional relevance for the plant. These topics are currently loosely included in the discussion and should be moved to the background information. Both the introduction and discussion sections require careful restructuring, as they presently consist of loose statements lacking flow, structure, and depth. The results section needs more detail on the outcomes of the statistical tests, and percentages should not be used for comparison alone; references should be made to the direct results found in your experiments. More details can be found in the attached PDF.

Annotated reviews are not available for download in order to protect the identity of reviewers who chose to remain anonymous.

---

## Round 0.2 · Minor Revisions

I appreciate your efforts to improve your manuscript. You have improved it by correcting many of the points discussed. However, the conclusion section is not written properly. It is not appropriate to mention the findings of previous studies at the beginning of the conclusion section. Lines 259-260 should be moved to the appropriate place in the discussion section and the literature should be cited. The conclusion section should briefly mention the results of your study and include recommendations for future studies.

Reviewer 2 ·

Basic reporting

The authors made significant changes in the manuscript.

Experimental design

see section 1

Validity of the findings

see section 1

Additional comments

see section 1

---

## Round 0.3 · accepted · Accept

I am pleased to announce the acceptance of your manuscript.